# RIPK2 induces docetaxel resistance in prostate cancer through the NF-κB/P-gp signaling pathway

**Shaoqiang Xing**[1,2☯], **Zhaoliang Xu**[1,2☯], **Sheng Zeng**[2], **Minghao Yue**[2], **Wenzhou Xing**[2], **Qian Liu**[1,2]*

**1** The First Central Clinical School, Tianjin Medical University, Tianjin, China, **2** Department of Urology, Tianjin First Central Hospital, Tianjin, China

☯ These authors contributed equally to this work.
* simonlq1971@126.com

## Abstract

Chemoresistance is a reason for treatment failure in prostate cancer. Receptor-interacting protein kinase 2 (RIPK2) has been shown to play a role in drug resistance in various cancers; however, its role and the underlying mechanism of chemoresistance in prostate cancer are unclear. We analyzed data from The Cancer Genome Atlas for RIPK2 expression in prostate cancer and its association with clinicopathological features. We also elucidated the role and mechanism of action of RIPK2 in prostate cancer cell resistance to docetaxel (DTX). The results showed that RIPK2 expression was upregulated in prostate cancer tissues and was associated with poor pathological grading. RIPK2 was also upregulated in 22RV1/DTX, C4-2/DTX, PC-3/DTX, and DU145/DTX cell lines and involved in DTX resistance. Mechanistic experiments revealed that RIPK2 was involved in DTX resistance by upregulating P-glycoprotein (P-gp) expression through the activation of the NF-κB signaling pathway. Xenograft tumor experiments confirmed that inhibition of RIPK2 or P-gp enhanced the efficacy of DTX in suppressing PC-3/DTX growth. Taken together, these results suggest that RIPK2 mediates DTX resistance in prostate cancer cells through the NF-κB/P-gp signaling pathway. RIPK2 and its downstream signaling molecules are potential targets for the treatment of chemoresistant prostate cancer.

## Introduction

Prostate cancer is a common genitourinary tumor in men with serious effects on patients' physical and mental health. Between 2015 and 2022, the incidence of prostate cancer in China increased from 10.2/100,000 to 18.6/100,000 [1,2]. Currently, treatments for prostate cancer include surgery, endocrine therapy, radiation therapy, chemotherapy, and immunotherapy [3,4]. Chemotherapy is an important strategy for the treatment of advanced metastatic prostate cancer following endocrine or

**Data availability statement:** All relevant data are within the manuscript and its Supporting information files.

**Funding:** This work was supported by the Tianjin Key Medical Discipline (Specialty) Construction Project (TJYXZDXK-3-024C). The funders had no role in study design, data collection and analysis, decision to publish, or preparation of the manuscript.

**Competing interests:** The authors have declared that no competing interests exist.

radiation therapy failure, and the use of drugs such as docetaxel (DTX), carboplatin, and olaparib has significantly prolonged patient survival [5,6]. However, tumors tend to develop acquired resistance to these drugs when administered over long periods, leading to treatment failure. Therefore, overcoming chemoresistance in prostate cancer is of great clinical importance.

Receptor-interacting protein kinase 2 (RIPK2) is a serine/threonine/tyrosine kinase that was first reported to be involved in inflammation and innate immunity [7,8]. Recent studies have shown that RIPK2 also plays pivotal roles in tumorigenesis and malignant tumor progression. Clinical studies have found that RIPK2 is associated with poor prognosis in pancreatic [9], lung [10], renal [11], and colorectal [12] cancers and can be used as a prognostic marker. RIPK2 is involved in tumor cell proliferation [13], migration [14], invasion [15], and metastasis [16]. In addition, RIPK2 may play a role in tumor immune escape by inhibiting antigen presentation and the cytotoxicity of cytotoxic T cells [17]. Recent studies have shown that RIPK2 mediates paclitaxel resistance in ovarian cancer [18] and temozolomide resistance in glioma [19]. Other studies have shown that RIPK2 promotes prostate cancer metastasis by increasing c-Myc activity via mitogen-activated protein kinase 7 [20]. However, whether RIPK2 mediates chemoresistance in prostate cancer remains unclear.

P-glycoprotein (P-gp) is an ATP-dependent carrier protein encoded by the human multidrug resistance gene *MDR1* [21]. P-gp is widely distributed in normal human tissues, and its expression effectively protects organisms from exogenous harmful substances under physiological conditions [21]. Under pathological conditions, P-gp may be involved in the development of chemoresistance in tumor cells [22]. However, the involvement of RIPK2 in prostate cancer chemoresistance and its association with P-gp has not yet been reported. In this study, we aimed to observe the effect of RIPK2 on the chemoresistance of prostate cancer cells and the mediatory role of P-gp to provide a reference for the development of novel therapeutics.

## Materials and methods

### Access to public datasets

Prostate cancer RIPK2 expression and Gleason scores from The Cancer Genome Atlas (TCGA) dataset were analyzed using the UALCAN portal (http://ualcan.path.uab.edu/), which analyzes cancer omics data and provides comprehensive transcriptomic data on cancer from TCGA. A total of 497 tumor samples and 52 normal tissue samples were included in the analysis. Gleason score grouping and subsequent statistical analysis were performed using the default parameters of UALCAN.

### PC-3/DTX resistant cell line construction

22RV1, C4-2, PC-3, and DU145 cells were purchased from the BeNa Culture Collection (Xinyang, China). Cells were cultured in F-12K medium (Gibco-BRL, Grand Island, NY, USA) containing 10% fetal bovine serum (FBS) (Gibco-BRL). 22RV1/DTX, C4-2/DTX, PC-3/DTX, and DU145/DTX DTX-resistant cell lines were obtained after six months by gradually increasing the concentration of DTX (MedChem Express, Monmouth Junction, NJ, USA) with intermittent induction.

### *In vitro* cell assays

The Cell Counting Kit-8 (CCK-8, MedChem Express) assay was performed as follows: cells were inoculated in 96-well plates overnight and treated with a DTX concentration gradient (0.001–100 nM) for 48 h. Ten microliters of CCK-8 solution were added to each well according to the manufacturer's instructions and incubated for 4 h. Absorbance was measured at 450 nm using an enzyme marker (Bio-Rad, Hercules, CA, USA). Crystalline violet staining was performed as follows: cells were fixed with 4% paraformaldehyde (Codow, Guangzhou, China) and treated with 0.5% crystal violet staining solution (Solarbio, Beijing, China) for 10 min. Cells were observed and photographed under a TS2 inverted microscope (Nikon, Tokyo, Japan).

### siRNA transfection

RIPK2 siRNA and control siRNA were designed and synthesized by Genomeditech (Shanghai, China). Cells were transfected using Lipofectamine 2000 (Biosynth, Tianjin, China) according to the manufacturer's instructions. The 150 pM of siRNA was used for transfection. After 8 h of transfection, the medium was replaced with F-12K medium containing 10% FBS. After 24 h of transfection, the cells were ready for use in subsequent experiments. siRNA complementary to RIPK2 mRNA with the following sequences: 5′-CACCAATCCTTTGCAGATAAT-3′, and nontargeting control siRNA, 5′-UUCUCCGAACGUGUCACGU-3′

### Plasmid transfection

The pLV-Puro RIPK2-containing plasmid and the empty plasmid were designed and synthesized by Genomeditech. Cells were transfected using Lipofectamine 2000 according to the manufacturer's instructions. 8 μg/ml of plasmid was used for transfection. After 8 h of transfection, the medium was replaced with F-12K medium containing 10% FBS. Cell lines with stable expression were screened using a monoclonal method.

### Western blotting

Protein quantification and sample preparation were performed using a bicinchoninic acid kit (Absin, Shanghai, China) and loading buffer (Absin), respectively. The total protein was separated using 10% sodium dodecyl sulfate-polyacrylamide gel electrophoresis and transferred onto a polyvinylidene difluoride (PVDF) membrane (LMAI, Shanghai, China). The PVDF membrane was closed in a sealing solution for 4 h, and then successively incubated with the primary and secondary antibodies. Primary antibodies included antibodies against RIPK2 (rabbit monoclonal antibody, 1:1000), GAPDH (rabbit monoclonal antibody, 1:2000), p-NF-κB p65 (rabbit monoclonal antibody, 1:1000), NF-κB p65 (rabbit monoclonal antibody, 1:2000), IκBα (rabbit monoclonal antibody, 1:1000), and P-gp (rabbit monoclonal antibody, 1:1000) (Cell Signaling Technology, Danvers, MA, USA). A sheep anti-rabbit IgG HRP peroxidase-conjugated antibody was used as the secondary antibody (1:4000) (Cell Signaling Technology). The PVDF membranes were treated with an enhanced chemiluminescence kit and placed in a gel imager (Bio-Rad) for imaging and photography.

### Immunofluorescence assay

Cells were inoculated into confocal dishes and cultured for 24 h. The cells were sequentially fixed, permeabilized, and closed using methanol, 0.1% Triton X-100, and 5% goat serum (Solarbio). Cells were incubated with NF-κB p65 (1:200) or P-gp (1:100) (Absin) antibodies. The primary antibodies were then conjugated to goat anti-rabbit IgG-AF488 and goat anti-mouse IgG-AF555 antibodies (Absin). Cells were blocked by incubation with 4',6-diamidino-2-phenylindole (DAPI, Solarbio) for nuclear staining followed by dropwise addition of anti-fluorescence burst blocking solution. Cells were imaged and photographed under a fluorescence microscope (Nikon).

## Flow cytometry

Cells were digested with ethylenediaminetetraacetic acid-free trypsin (Absin) and collected in 5 ml centrifuge tubes. Flow cytometry was performed using an apoptosis detection kit (Absin) according to the manufacturer's instructions. Briefly, cells were collected and resuspended with 300 μL of 1 × binding buffer. 5 μL Annexin V-FITC was added to the suspension and incubated for 15 min, followed by 5 μL PI for 2 min. Apoptosis was detected using an Accuri C6 Plus flow cytometer (BD Biosciences, San Jose, CA, USA).

## Animal experiments

The animal experiments were approved by the Ethics Committee of Tianjin First Central Hospital (Protocol Number: 2023-0815-001) and followed the Regulations on the Management of Experiments approved by the State Council of China. This study followed the ARRIVE guidelines. All surgery was performed under isoflurane anesthesia, and all efforts were made to minimize suffering. Six- to seven-week-old BALB/c-Nude mice were purchased from Changzhou Cavens Laboratory Animal Co. The experimental animals were kept in a SPF-level Laboratory Animal Room for one week, and then subcutaneously injected with 200 μL PC-3/DTX cell suspension ($1 \times 10^7$ cells/mL) in the right dorsum. Tumor volume was measured using Vernier calipers and calculated using the following formula: tumor volume (TV) = tumor long diameter × short diameter $^2$/2. Tumor measurement and analysis were performed using a blinded method. When the xenograft tumors grew to approximately 200 mm$^3$, the mice were randomly divided into four groups (4 animals per group): DTX (MedChem Express, Monmouth Junction, NJ, USA), DTX + GSK583 (MedChem Express), DTX+tariquidar (MedChem Express), and Control. DTX (15 mg/kg, i.v.), GSK583 (3 mg/kg, p.o.), and tariquidar (15 mg/kg, p.o.) were administered twice per week for four weeks. The control group received the corresponding vehicle via gavage and tail vein injection. Daily monitoring of animal health and behavior was conducted, with humane endpoints implemented when animals showed >15% weight loss (1-week period), excessively large tumors (≥20 mm in diameter), or overt signs of poor condition (e.g., reduced activity, anorexia). At the end of the experiment, mice were anesthetized with 3% isoflurane and subsequently rapidly sacrificed by carbon dioxide asphyxiation. The tumors were stripped, weighed, and subsequently placed in 4% paraformaldehyde for internal fixation.

## Immunohistochemistry

Tissues were cut into small pieces and placed in an embedding box for dehydration and tissue embedding. The embedded tissue was cut into 4 μm sections. Sections were spread in warm water at 37°C and then fixed on slides before baking. Sections were sequentially deparaffinized, hydrated, antigenically repaired (100℃, 15 min), and incubated with 5% goat serum for 30 min. The sections were then incubated overnight with antibodies against Ki67 (mouse monoclonal antibody; 1:50; Santa Cruz Biotechnology, Santa Cruz, CA, USA) and P-gp (1:100). Ki67 or P-gp antibody was conjugated with an enzymatically labeled goat anti-mouse/rabbit IgG secondary antibody (ZSGB-BIO, Beijing, China). They were then washed to remove any unbound antibodies and incubated with DAB chromogenic solution (ZSGB-BIO) until the color developed. Sections were re-stained with hematoxylin (ZSGB-BIO) for nuclei, dehydrated, and sealed with neutral gum (ZSGB-BIO). The following day, the sections were observed and photographed under an orthostatic microscope (Nikon). The integrated optical density values of tissue sections were measured using Image-Pro Plus 6.0 software (Media Cybernetics, Inc., Rockville, MD, USA).

## Statistical analysis

Statistical analyses were performed using SPSS software (version 27.0; IBM, Armonk, NY, USA) for Windows. In vitro data were based on biological triplicate experiments, with each biological replicate performed in at least three technical repeats to ensure reproducibility. All results are shown as the mean ± standard deviation. Data comparisons between

multiple groups were performed using one-way analysis of variance, and multiple comparisons were made using Tukey's test. $P<0.05$ was considered significant.

## Results

### RIPK2 expression in prostate cancer

We first analyzed prostate cancer RIPK2 expression in TCGA database using the University of Alabama at Birmingham cancer data analysis (UALCAN) portal. RIPK2 mRNA levels were upregulated in prostate cancer tissues when compared to those in normal prostate tissues (Fig 1A). RIPK2 expression was consistently positively correlated with prostate cancer Gleason scores (Fig 1B).

### RIPK2 induces DTX resistance in prostate cancer cells

We induced DTX-resistant prostate cancer cell lines 22RV1/DTX, C4-2/DTX, PC-3/DTX, and DU145/DTX by progressively increasing the drug concentration, with resistance indices of 16.2, 6.1, 11.5, and 6.6, respectively (Fig 2A). We further found that RIPK2 protein expression was upregulated in these four cells (Fig 2B). To observe the relationship between RIPK2 protein and DTX resistance, we used siRNA to interfere with RIPK2 expression and observed the changes in sensitivity to DTX in 22RV1/DTX, C4-2/DTX, PC-3/DTX, and DU145/DTX cells. The results showed that each DTX-resistant cell showed increased sensitivity to DTX after interfering with RIPK2 protein expression, with PC-3/DTX showing the most pronounced increase (Fig 2C). For this reason, we used PC-3/DTX cells for subsequent studies. Verification of the silencing effect of RIPK2 by crystal violet staining revealed that silencing of RIPK2 enhanced the sensitivity of PC-3/DTX cells to DTX (Fig 2D). In addition, we overexpressed RIPK2 in PC-3 cells and found that the sensitivity of the cells to DTX was significantly reduced (Fig 2E-2G).

### RIPK2 induces activation of the NF-κB signaling pathway in prostate cancer cells

To explore the biological role of RIPK2 in prostate cancer cells, we examined the NF-κB signaling pathway. We found that transfection of the RIPK2 plasmid induced NF-κB p65 phosphorylation, as well as downregulation of IκBα protein

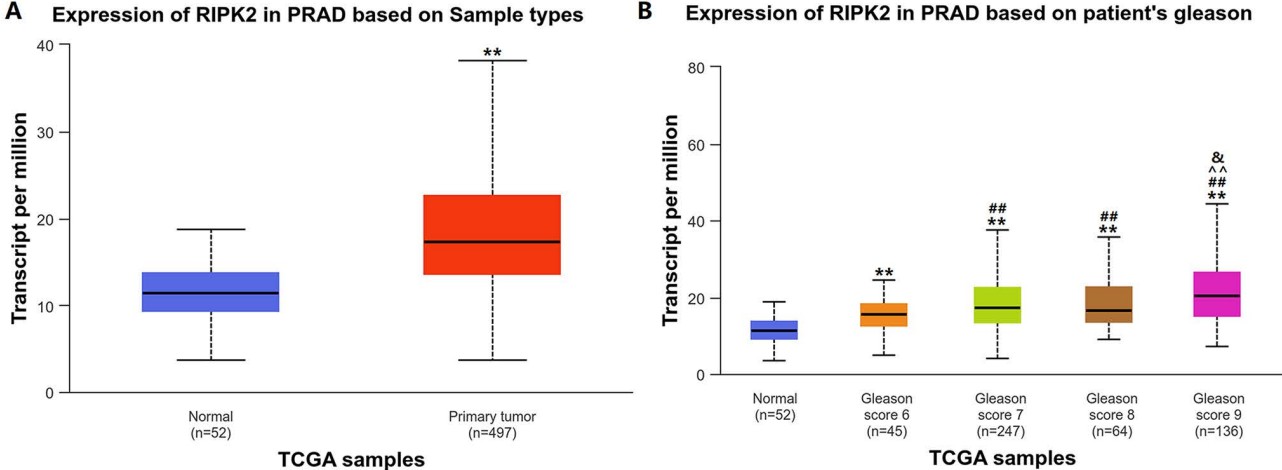

**Fig 1. RIPK2 expression in prostate cancer cells. (A)** RIPK2 mRNA expression levels in normal prostate and prostate cancer tissues as identified by TCGA data. **P<0.01 compared to Normal. **(B)** Association of RIPK2 mRNA expression with Gleason scores in prostate cancer based on TGCA data. ** P<0.01 compared to Normal; ## P<0.01 compared to Gleason score 6; ^^ P<0.01 compared to Gleason score 7; & P<0.05 compared to Gleason score 8.

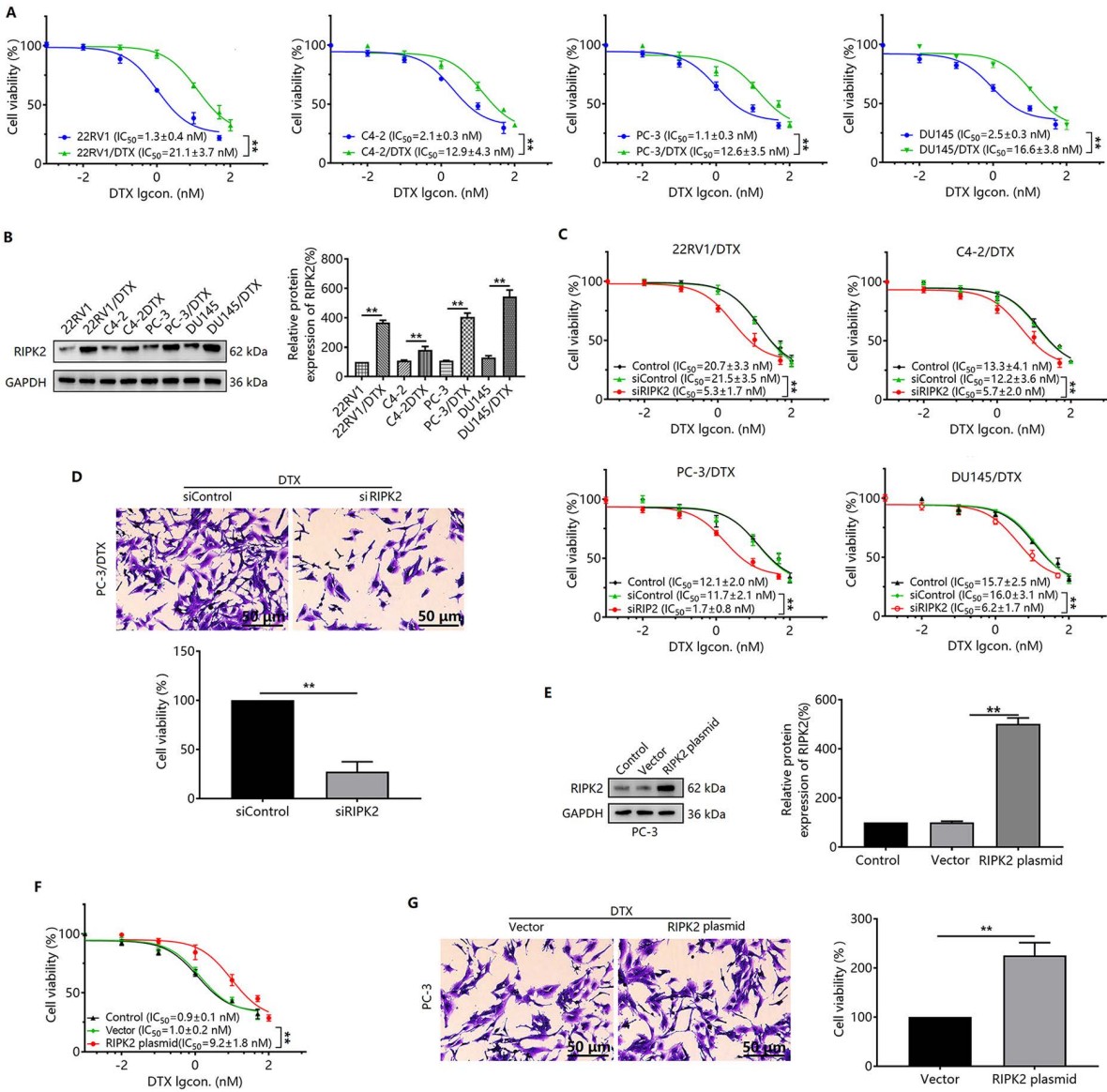

**Fig 2. Effects of RIPK2 on DTX resistance in prostate cancer cells. (A)** 22RV1/DTX, C4-2/DTX, PC-3/DTX, and DU145/DTX cells were treated with DTX (0.001~100 nM) for 48 **h.** Cell viability was measured by CCK-8 assay. **(B)** Western blotting was performed to detect the expression level of RIPK2 in normal and drug-resistant prostate cancer cells. **(C)** Effect of DTX on 22RV1/DTX, C4-2/DTX, PC-3/DTX, and DU145/DTX cell viability after siRNA interference with RIPK2 expression. **(D)** Crystalline violet staining assay to detect the sensitivity of PC-3/DTX cells to DTX after RIPK2 silencing. **(E)** RIPK2 protein expression level in PC-3 cells after transfection with RIPK2 plasmid. RIPK2 expression levels in PC-3 cells after transfection with the RIPK2 plasmid. Sensitivity of PC-3 cells to DTX was detected by CCK-8 assay **(F)** and crystal violet staining **(G)** after overexpression of RIPK2. $^{**}P < 0.01$. n = 3 in A-G.

expression in PC-3 cells (Fig 3A). Transfection of the RIPK2 plasmid also induced the nuclear translocation of NF-κB p65 (Fig 3B). The addition of siRNAs against RIPK2 led to decreased NF-κB p65 phosphorylation levels and upregulated IκBα protein expression in PC-3/DTX cells (Fig 3C). Meanwhile, the expression of NF-κB p65 in the nuclei of PC-3/DTX cells decreased after RIPK2 knockdown (Fig 3D).

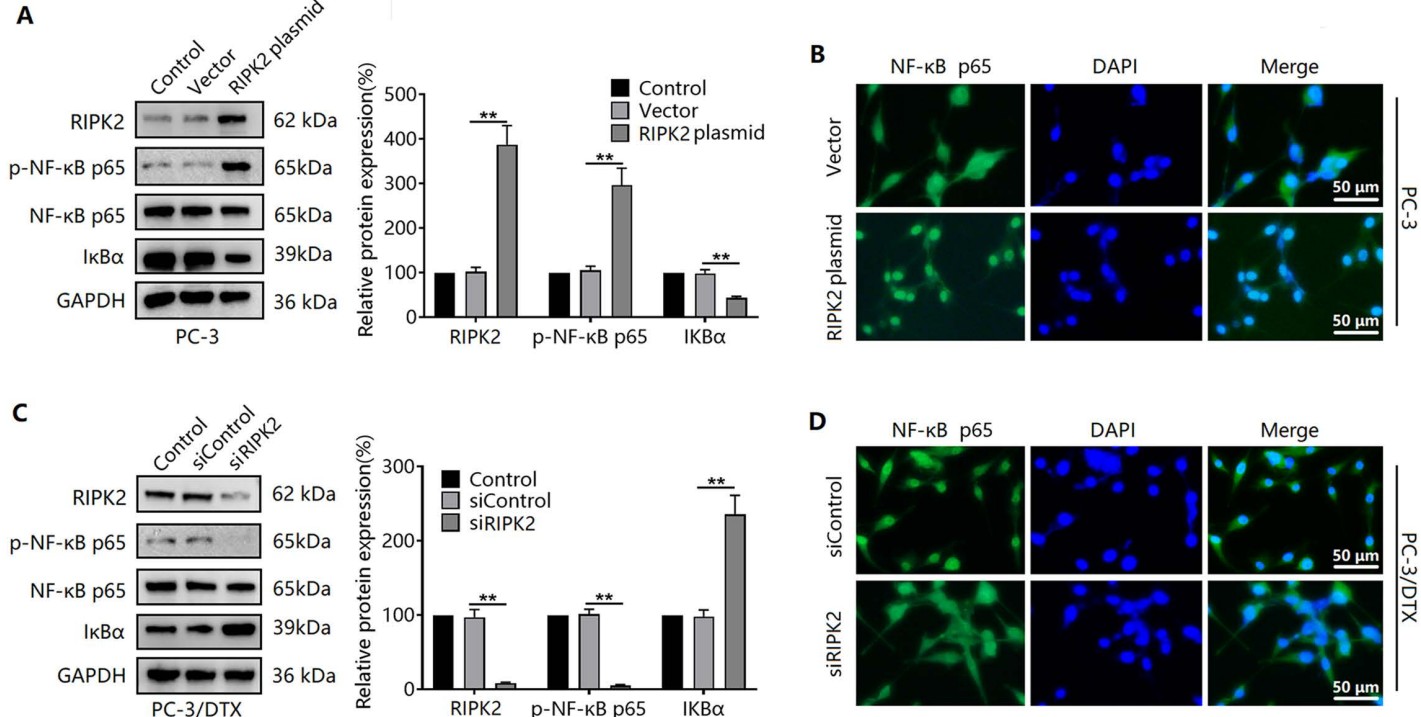

**Fig 3. Effects of varying RIPK2 expression on the NF-κB signaling pathway. (A)** Changes in p-NF-κB p65 and IκBα expression in PC-3 cells after overexpression of RIPK2. **(B)** Changes in the subcellular distribution of NF-κB p65 in PC-3 cells after RIPK2 overexpression. **(C)** Changes in p-NF-κB p65 and IκBα expression in PC-3/DTX cells after RIPK2 knockdown. **(D)** Changes in NF-κB p65 subcellular distribution in PC-3/DTX cells after RIPK2 knockdown. $^{**}P < 0.01$. n = 3 in A-D.

## RIPK2/NF-κB signaling mediates the upregulation of P-gp protein expression

We further found that RIPK2 overexpression in PC-3 cells was accompanied by an upregulation in P-gp protein expression (Fig 4A). Meanwhile, silencing RIPK2 expression resulted in the downregulation of P-gp expression in PC-3/DTX cells (Fig 4A). Immunofluorescence experiments showed that P-gp expression and the nuclear translocation of NF-κB p65 increased in RIPK2-overexpressing PC-3 cells (Fig 4B). Silencing of RIPK2 had the opposite effect in PC-3/DTX cells (Fig 4B). To verify the relationship between RIPK2/NF-κB signaling and P-gp protein expression, we assessed the effects of NF-κB inhibitors (JSH-23, BAY11−7085, and SN50) on PC-3 cells. Pretreatment with JSH-23, BAY11−7085, or SN50 inhibited upregulation of P-gp expression induced by RIPK2 overexpression (Fig 4C), thus leading to decreased P-gp expression in PC-3/DTX cells (Fig 4D).

## Inhibition of the RIPK2/NF-κB/P-gp signaling pathway enhances the sensitivity of prostate cancer cells to DTX

We further validated the role of the RIPK2/NF-κB/P-gp signaling pathway in the resistance of prostate cancer cells to DTX. We found that treatment with either SN50, an NF-κB inhibitor, or tariquidar, a P-gp inhibitor, enhanced the inhibitory effect of DTX on the proliferation of RIPK2-overexpressing PC-3 cells (Fig 5A). Treatment with SN50 or tariquidar also enhanced DTX-induced apoptosis in RIPK2-overexpressing PC-3 cells (Fig 5B). We confirmed the role of the RIPK2/NF-κB/P-gp signaling pathway in PC-3/DTX cells. Treatment with SN50 or tariquidar had similar effects on proliferation and apoptosis in PC-3/DTX cells (Fig 5C and 5D). In addition, treatment with GSK583, a RIPK2 inhibitor, enhanced the efficacy of DTX in PC-3/DTX cells, leading to decreased proliferation and increased apoptosis (Fig 5C and 5D).

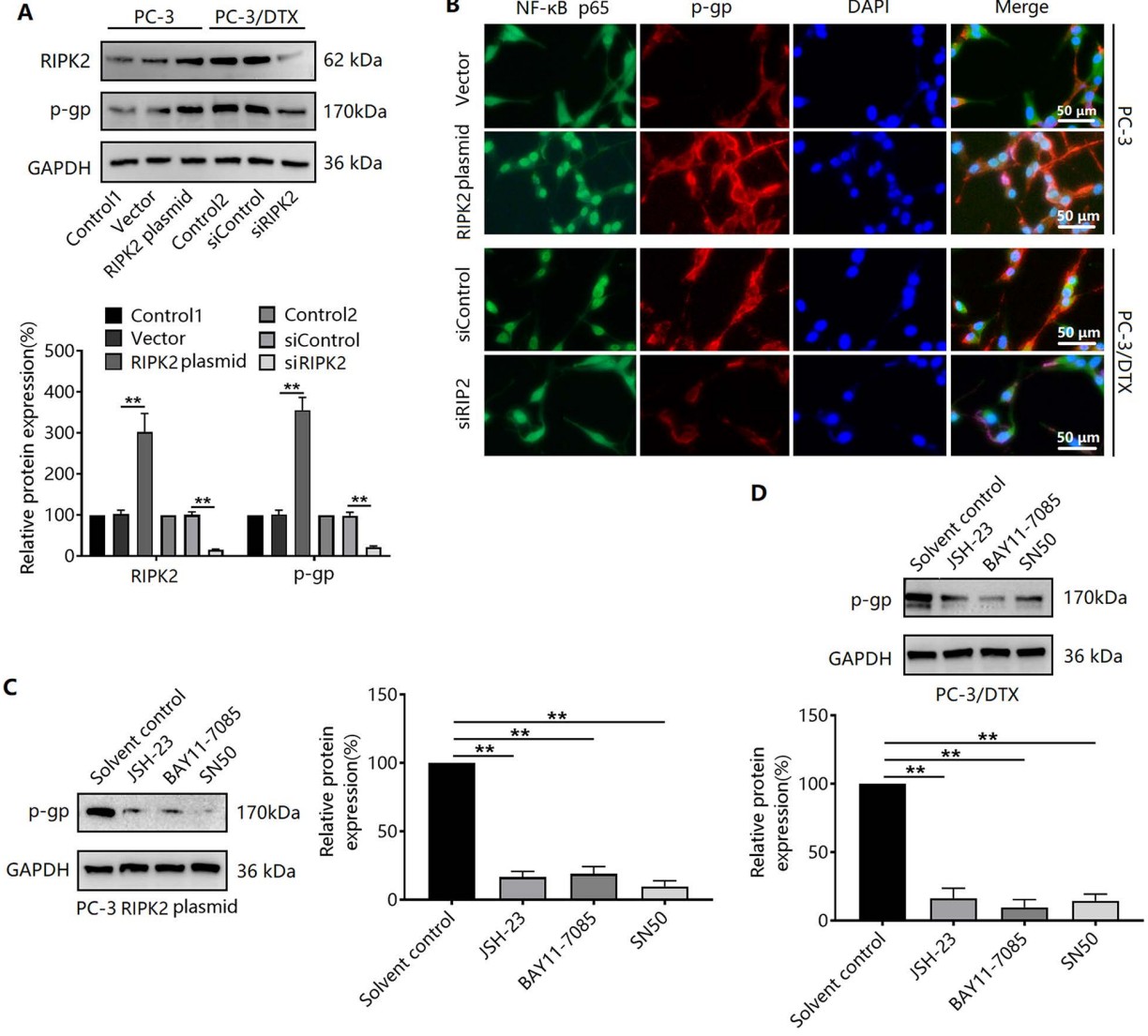

**Fig 4. Effect of the RIPK2/NF-κB signaling pathway on P-gp expression. (A)** Effects of exogenous regulation of RIPK2 expression in PC-3 and PC-3/DTX cells and western blotting for P-gp expression. **(B)** Effect of RIPK2 on subcellular localization of NF-κB p65 and P-gp expression, as determined by immunofluorescence. PC-3/DTX cells **(C)** or RIPK2-overexpressing PC-3 cells **(D)** were treated with the NF-κB inhibitors JSH-23, BAY11-7085, or SN50, and P-gp expression was detected using western blotting. **$P < 0.01$. n = 3 in A-D.

The results of the animal experiments showed that GSK583 treatment downregulated P-gp expression in PC-3/DTX xenograft tumor tissues (Fig 6A). Treatment with GSK583 or tariquidar in combination with DTX downregulated the expression of the proliferation marker Ki67 (Fig 6A).

## Inhibition of the RIPK2/P-gp signaling pathway enhances DTX sensitivity in PC-3/DTX xenograft tumors

Finally, we validated the role of the RIPK2/P-gp signaling pathway in DTX resistance using xenograft tumor experiments. The experimental results showed that DTX did not have a marked tumor-suppressing effect on PC-3/DTX xenograft tumors (Fig 6B-6D). GSK583 and tariquidar enhanced the sensitivity of PC-3/DTX xenograft tumors to DTX (Fig 6B-6D).

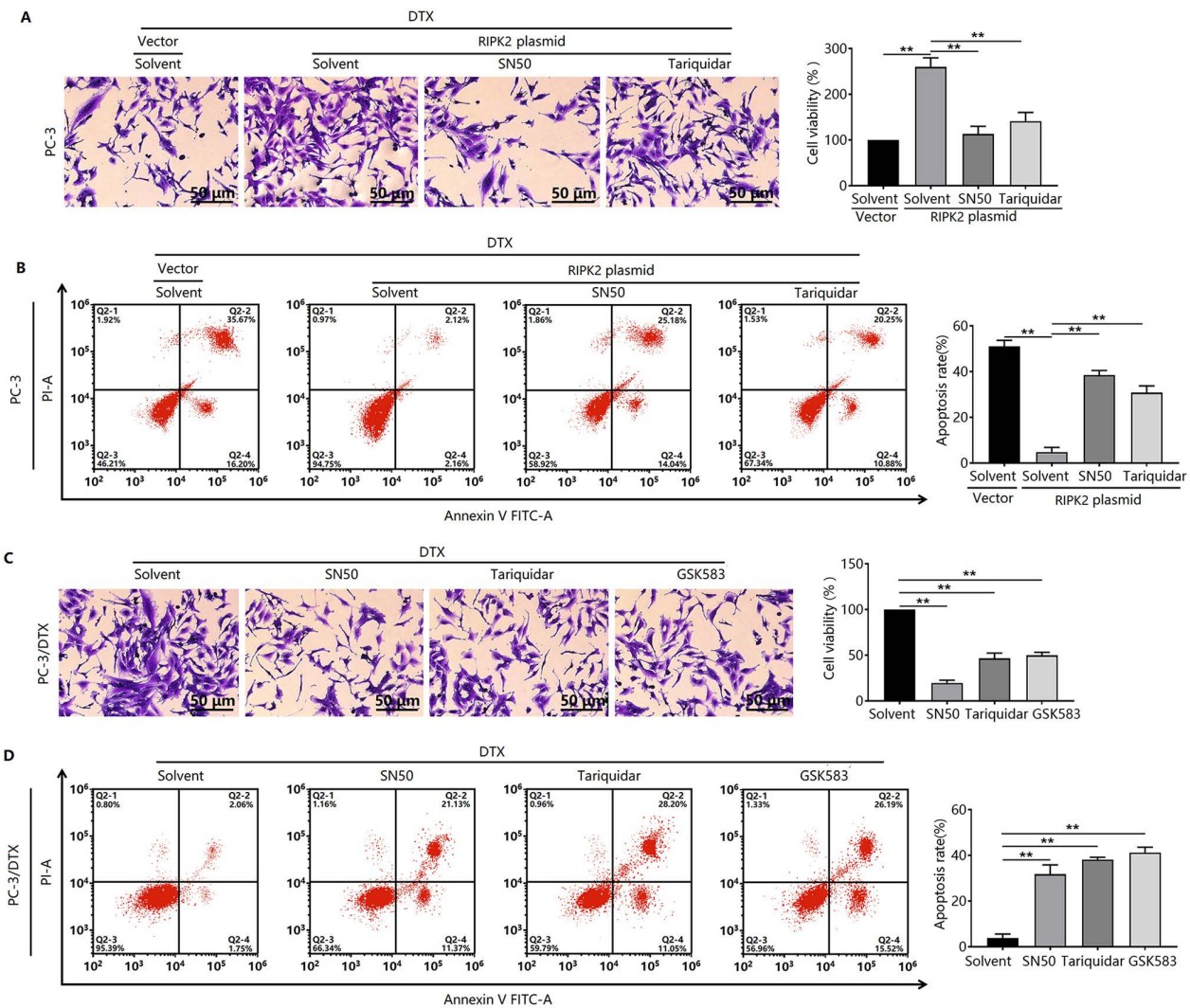

**Fig 5. Inhibition of the RIPK2/NF-κB/P-gp signaling pathway enhances the sensitivity of prostate cancer cells to DTX.** RIPK2-overexpressing PC-3 cells were treated with the NF-κB inhibitor SN50 or P-gp inhibitor tariquidar, and cell sensitivity to DTX (12 nM) was assessed using crystal violet staining (**A**) and flow cytometry (**B**). PC-3/DTX cells were treated with SN50, tariquidar, or the RIPK2 inhibitor GSK583, and cell sensitivity to DTX (12 nM) was detected using crystal violet staining (**C**) and flow cytometry (**D**). ***P* < 0.01. n = 3 in A-D.

## Discussion

RIPK2 plays a vital role in the progression of various tumors, including prostate cancer. Studies have shown that *Serratia marcescens* promotes esophageal squamous cell carcinoma xenograft tumor growth through the RIPK2-mediated activation of NF-κB [23]. RIPK2 is also involved in the regulation of gastric cancer tumor growth by inhibiting apoptosis [24]. The paired protein kinases PRKCI and RIPK2 promote pancreatic cancer growth and metastasis through activation of the JNK/ERK pathway via NF-κB phosphorylation [16]. Yan et al. [20] reported that RIPK2 is amplified in approximately 65% of metastatic prostate cancers, and its expression is associated with disease progression and poor prognosis. Our TCGA analysis revealed that RIPK2 mRNA levels were upregulated in

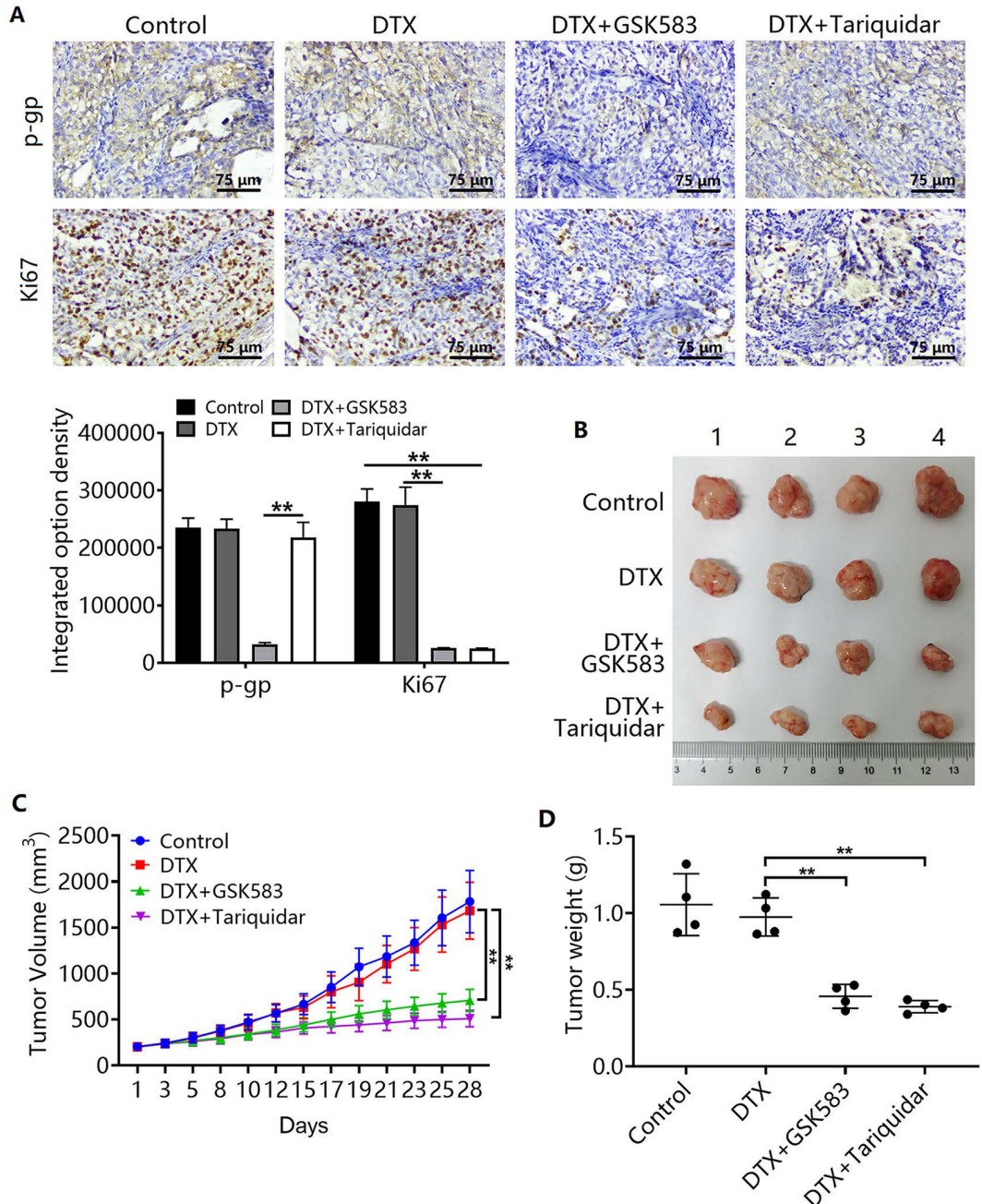

**Fig 6. Effects of RIPK2 and P-gp inhibition on DTX-treated prostate cancer xenografts. (A)** P-gp and Ki67 expression in xenograft tumor tissues of the control, DTX, DTX+GSK583, and DTX+tariquidar groups. $^{**}P<0.01$. n = 3 in A. **(B)** Photographs of xenograft tumors in the control, DTX, DTX+GSK583, and DTX+tariquidar groups after 28 days of treatment. **(C)** Comparison of tumor volumes by group. **(D)** Comparison of tumor weights by group. $^{**}P<0.01$. n = 4 in B-D.

prostate cancer tissues. Consistently, RIPK2 expression positively correlated with prostate cancer Gleason scores. These results agree with those of previous studies. RIPK2 is associated with drug resistance in malignant tumors. RIPK2 is highly expressed in paclitaxel-resistant ovarian cancer cell lines, and high RIPK2 expression is associated with paclitaxel resistance in plasmacytoid ovarian cancer [18]. Pan-cancer analyses revealed that RIPK2 exacerbates cytotoxic T-lymphocyte dysfunction and promotes resistance to immunotherapies through the JAK/STAT3 signaling pathway and γ-interferon response [25]. Furthermore, RIPK2 contributes to temozolomide resistance by inducing stemness in glioma cells through the NF-κB signaling pathway [19]. These findings indicate that RIPK2 plays a role in chemotherapy and immunotherapy resistance in a variety of malignant tumors. Our study found that RIPK2 expression was elevated in 22RV1/DTX, C4-2/DTX, PC-3/DTX, and DU145/DTX prostate cancer cells, and silencing RIPK2 expression increased cell sensitivity to DTX. Additionally, RIPK2-overexpressing PC-3 cells showed decreased DTX sensitivity. These results suggested RIPK2 involvement in the resistance of prostate cancer cells to DTX.

NF-κB is part of a regulatory pathway involved in inflammatory response, and dysregulated NF-κB activity can lead to inflammation-related diseases and cancer; thus, it is a potential target for cancer therapy [26,27]. The NF-κB signaling pathway is downstream of RIP2, thereby mediating RIP2-induced cancer progression [28,29]. Thus, we examined the effects of varying RIPK2 expression in prostate cancer cells on the NF-κB signaling pathway. We found that RIPK2 overexpression induced upregulation of NF-κB P65 phosphorylation and nuclear translocation in PC-3 cells while downregulating the expression of the NF-κB-inhibiting protein IκBα. In contrast, silencing RIPK2 expression in PC-3/DTX cells reduced NF-κB P65 phosphorylation and nuclear translocation but upregulated IκBα expression. This suggests that RIPK2 activates the NF-κB signaling pathway in prostate cancer cells.

We further found that exogenous regulation of RIPK2 expression positively regulated P-gp expression. Additionally, NF-κB inhibition reversed the RIPK2-induced upregulation of P-gp expression in prostate cancer cells. It is suggested that RIPK2 regulates P-gp protein expression in prostate cancer cells through the NF-κB pathway. P-gp, an ATP-dependent transmembrane protein, has a large central pathway for drug binding and efflux into its transmembrane region [30,31]. Upregulation of P-gp expression in cancer cells enhances the efflux of chemotherapeutic drugs, thereby reducing drug accumulation in cancer cells and ultimately producing chemoresistance [32,33]. N-methylpretrichodermamide B enhances DTX sensitivity in drug-resistant prostate cancer cells by downregulating P-gp expression [34]. Spongian diterpenes demonstrated strong anti-proliferative effects in human DTX-resistant prostate cancer cells, and this effect was mainly attributed to the inhibition of P-gp-mediated drug efflux [35]. These results suggest that RIPK2/NF-κB-induced DTX resistance in our study may be associated with upregulated P-gp expression. We further validated the role of the RIPK2/NF-κB/P-gp pathway in DTX resistance using *in vitro* cellular experiments and xenograft tumor experiments. We found that inhibition of either RIPK2, NF-κB, or P-gp enhanced the sensitivity of PC-3/DTX and RIPK2-overexpressing PC-3 cells to DTX. Combining the RIPK2 inhibitor GSK583 or P-gp inhibitor tariquidar with DTX inhibited Ki67 expression (a marker of proliferation) and enhanced the growth-inhibiting effect of DTX on PC-3/DTX xenograft tumors. In addition, GSK583 inhibited P-gp expression in PC-3/DTX xenograft tumors. These results suggest that RIPK2 is involved in DTX resistance by upregulating P-gp expression through activation of the NF-κB signaling pathway.

In conclusion, RIPK2 expression is upregulated in prostate cancer, and its expression correlates with pathological grading. Our study revealed a potential mechanism by which RIPK2 regulates the resistance of prostate cancer cells to chemotherapy. RIPK2 activates the NF-κB pathway and upregulates P-gp protein expression, which in turn mediates DTX resistance (Fig 7). Thus, RIPK2 and its downstream signaling targets may serve as therapeutic targets for chemoresistant prostate cancers. However, the biological mechanism of action of RIPK2 in prostate cancer is not fully understood and requires further investigation.

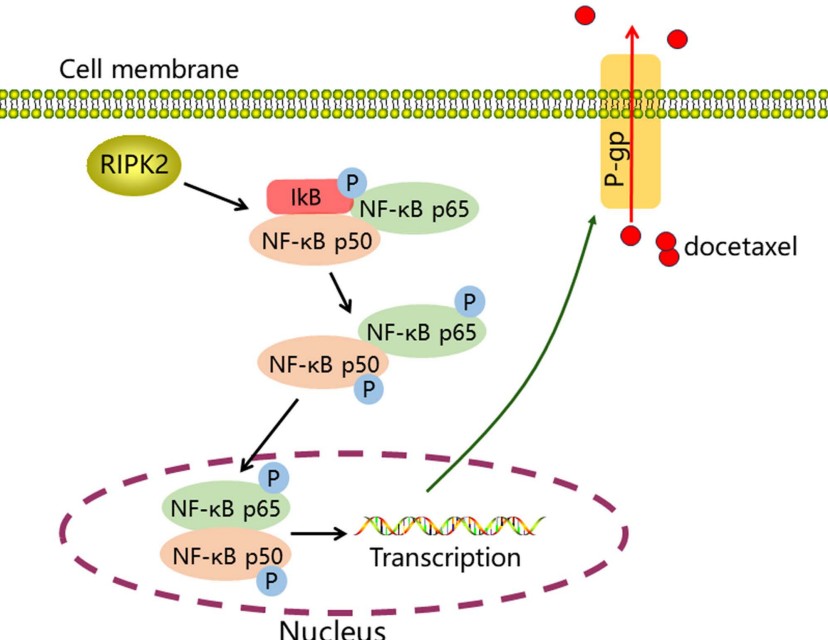

**Fig 7. Schematic diagram of the mechanism underlying RIPK2 inducing DTX resistance.** RIPK2 activates the NF-κB pathway and upregulates P-gp protein expression, which in turn mediates DTX resistance.

## Supporting information

**S1 File. Raw images.**
(PDF)

**S2 File. Dataset supporting the experimental results of this study.**
(PDF)

## Author contributions

**Formal analysis:** Zhaoliang Xu.

**Funding acquisition:** Qian Liu.

**Investigation:** Shaoqiang Xing, Zhaoliang Xu, Sheng Zeng, Minghao Yue, Wenzhou Xing.

**Methodology:** Shaoqiang Xing, Zhaoliang Xu, Sheng Zeng, Minghao Yue, Wenzhou Xing.

**Project administration:** Zhaoliang Xu.

**Software:** Shaoqiang Xing.

**Supervision:** Qian Liu.

**Validation:** Minghao Yue.

**Visualization:** Sheng Zeng.

**Writing – original draft:** Shaoqiang Xing.

**Writing – review & editing:** Qian Liu.

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
