## [Decision Letter · Decision Letter 0]

15 Oct 2025

Dear Dr. Liu,

Thank you for submitting your manuscript to PLOS ONE. After careful consideration, we feel that it has merit but does not fully meet PLOS ONE’s publication criteria as it currently stands. Therefore, we invite you to submit a revised version of the manuscript that addresses the points raised during the review process.

We look forward to receiving your revised manuscript.

Kind regards,

Saki Raheem, PhD

Academic Editor

PLOS ONE

Journal Requirements:

“This work was supported by the Tianjin Key Medical Discipline(Specialty) Construction Project.”

Additional Editor Comments:

The manuscript investigates the role of RIPK2 in mediating docetaxel (DTX) resistance in prostate cancer, proposing a mechanism via NF-kB activation and upregulation of P-glycoprotein (P-gp). The study includes in vitro experiments, TCGA dataset analysis, and xenograft models. The topic is relevant to therapeutic resistance in prostate cancer and fits PLOS ONE's emphasis on scientific rigour. However, several clarifications and improvements are needed before the manuscript can be considered for publication.

1. TCGA/UALCAN analysis- clarification needed:

The use of the UALCAN portal for TCGA analysis is appropriate, but a few reporting details would improve transparency and reproducibility. In particular, please indicate: How many tumour and normal samples were included? What statistical test did UALCAN apply, and how were the Gleason score groups defined?

If default UALCAN settings were used, a brief statement to that effect would be sufficient.

2. Please indicate whether in vitro data (e.g., IC50) were based on biological triplicates, and how many replicates were performed.

3. The in vivo experiments are generally well described, but to meet PLOS ONE and ARRIVE reporting standards, a few brief clarifications would improve transparency and reproducibility.

Please specify the number of animals per group in the MethodsIndicate whether randomisation was used when assigning mice to treatment groups.State whether blinding was applied during tumour measurement or analysis (or clarify if not performed).Briefly note the monitoring frequency and any humane endpoint criteria used before the scheduled sacrifice.clarify the nature of the control treatment/vehicle given to the drug-control group.

These additions would align the manuscript with ARRIVE  expectations without requiring new experiments.

4. Writing and Style: Minor language edits would improve clarity and consistency. There are occasional tense shifts and missing articles. The introduction is also quite long and could be streamlined to avoid repetition. A light language edit would help polish the text.

Reviewers' comments:

Reviewer's Responses to Questions

**Comments to the Author**

1. Is the manuscript technically sound, and do the data support the conclusions?

Reviewer #1: Yes

2. Has the statistical analysis been performed appropriately and rigorously?

Reviewer #1: Yes

3. Have the authors made all data underlying the findings in their manuscript fully available?

Reviewer #1: Yes

4. Is the manuscript presented in an intelligible fashion and written in standard English?

Reviewer #1: Yes

Reviewer #1: I am writing to provide feedback on the manuscript titled " RIPK2 induces docetaxel resistance in prostate cancer through the NF-κB/P-gp signaling pathway " (PONE-D-25-24339) which I understand is under consideration for publication. After a thorough review, I have identified several areas that could benefit from further attention to enhance the overall quality and impact of the paper.

References and Citations: It is crucial to ensure uniformity in the formatting of all references and citations, adhering to the required style guide. Consistency in this aspect upholds the manuscript's professionalism.

Clarity and Conciseness: The manuscript would benefit from a focus on clear and concise communication. Avoiding excessive jargon and simplifying complex ideas will make the paper more accessible to a broader audience.

Data Presentation: I recommend a review of how figures and tables are integrated into the paper. They should be clearly labeled and directly referenced in the text, ensuring they effectively support the paper's conclusions.

Logical Flow: Ensuring a coherent flow throughout the paper, with seamless transitions between sections, is key to keeping the reader engaged and ensuring the paper is easily understandable.

Abstract and Conclusion: These sections should succinctly summarize the main findings and their significance. It is important to avoid overstating results or generalizations not supported by the data.

Consistency in Terminology: Uniform use of technical terms throughout the manuscript is essential for clarity and to avoid potential confusion among readers.

Objective Language: Maintaining an objective tone throughout the paper is crucial. Subjective or evaluative language should be reserved for discussing hypotheses or theoretical frameworks.

Compliance with Journal Guidelines: Finally, ensuring that the manuscript aligns with the journal's specific guidelines in terms of formatting, submission, and ethics will facilitate a smoother review and publication process.

I trust these suggestions will be valuable in enhancing the manuscript's effectiveness and suitability for publication. Thank you for the opportunity to review this work.

Sincerely,

**Do you want your identity to be public for this peer review?** For information about this choice, including consent withdrawal, please see our Privacy Policy

Reviewer #1: No

While revising your submission, please upload your figure files to the Preflight Analysis and Conversion Engine (PACE) digital diagnostic tool, https://pacev2.apexcovantage.com/ . PACE helps ensure that figures meet PLOS requirements. To use PACE, you must first register as a user. Registration is free. Then, login and navigate to the UPLOAD tab, where you will find detailed instructions on how to use the tool. If you encounter any issues or have any questions when using PACE, please email PLOS atfigures@plos.org

---

## [Author Response · Author response to Decision Letter 1]

6 Nov 2025

2025/11/05

Dear Editor and Reviewers:

Thank you very much for your time and constructive comments on our manuscript entitled " RIPK2 induces docetaxel resistance in prostate cancer through the NF-κB/P-gp signaling pathway". We greatly appreciate the valuable suggestions, which have helped us significantly improve the quality of this work. According to the reviewers' comments, we have carefully revised the manuscript and addressed all the concerns raised. The main revisions are summarized as follows, and detailed responses to each comment are provided below. We hope the revised version meets the publication standards of PLOS ONE.

Sincerely,

Qian Liu

Department of Urology, First Central Clinical College, Tianjin Medical University, No. 24, Fukang Road, Nankai District, Tianjin, 300190, China. +86 18963106651

simonlq1971@126.com

Journal Requirements:

Reply: We have reviewed the manuscript according to the formatting requirements and file naming conventions specified by PLOS ONE.

“This work was supported by the Tianjin Key Medical Discipline(Specialty) Construction Project.”

Reply: We have incorporated the revised statement of the funding agency's role into the cover letter.

Reply: The blot/gel image data are included in the Supplementary Materials. We have supplemented the relevant information in the cover letter.

Additional Editor Comments:

The manuscript investigates the role of RIPK2 in mediating docetaxel (DTX) resistance in prostate cancer, proposing a mechanism via NF-kB activation and upregulation of P-glycoprotein (P-gp). The study includes in vitro experiments, TCGA dataset analysis, and xenograft models. The topic is relevant to therapeutic resistance in prostate cancer and fits PLOS ONE's emphasis on scientific rigour. However, several clarifications and improvements are needed before the manuscript can be considered for publication.

1. TCGA/UALCAN analysis- clarification needed:

The use of the UALCAN portal for TCGA analysis is appropriate, but a few reporting details would improve transparency and reproducibility. In particular, please indicate: How many tumour and normal samples were included? What statistical test did UALCAN apply, and how were the Gleason score groups defined?

If default UALCAN settings were used, a brief statement to that effect would be sufficient.

Reply: We have incorporated the number of cases used in the TCGA/UALCAN analysis into the ‘Materials and Methods’ section. Additionally, we performed Gleason score grouping and statistical analysis using default settings, with supplementary descriptions provided in the manuscript.

2. Please indicate whether in vitro data (e.g., IC50) were based on biological triplicates, and how many replicates were performed.

Reply: In vitro data (e.g., IC50 values) were based on biological triplicate experiments, with each biological replicate performed in at least three technical repeats to ensure reproducibility. We have supplemented the relevant descriptions in the manuscript.

3. The in vivo experiments are generally well described, but to meet PLOS ONE and ARRIVE reporting standards, a few brief clarifications would improve transparency and reproducibility.

Please specify the number of animals per group in the Methods

Indicate whether randomisation was used when assigning mice to treatment groups.

State whether blinding was applied during tumour measurement or analysis (or clarify if not performed).

Briefly note the monitoring frequency and any humane endpoint criteria used before the scheduled sacrifice.

clarify the nature of the control treatment/vehicle given to the drug-control group.

These additions would align the manuscript with ARRIVE expectations without requiring new experiments.

Reply: We have supplemented the relevant descriptions in the manuscript.

4. Writing and Style: Minor language edits would improve clarity and consistency. There are occasional tense shifts and missing articles. The introduction is also quite long and could be streamlined to avoid repetition. A light language edit would help polish the text.

Reply: We have invited native English speakers with academic writing experience to polish the manuscript, focusing on optimizing language fluency, correcting grammatical errors, and refining academic expression to meet the journal’s standards.

Reviewers' comments:

5. Review Comments to the Author

Reviewer #1: I am writing to provide feedback on the manuscript titled " RIPK2 induces docetaxel resistance in prostate cancer through the NF-κB/P-gp signaling pathway " (PONE-D-25-24339) which I understand is under consideration for publication. After a thorough review, I have identified several areas that could benefit from further attention to enhance the overall quality and impact of the paper.

References and Citations: It is crucial to ensure uniformity in the formatting of all references and citations, adhering to the required style guide. Consistency in this aspect upholds the manuscript's professionalism.

Reply: We have verified the references and citations.

Clarity and Conciseness: The manuscript would benefit from a focus on clear and concise communication. Avoiding excessive jargon and simplifying complex ideas will make the paper more accessible to a broader audience.

Reply: We have invited native English speakers with academic writing experience to polish the manuscript, focusing on optimizing language fluency, correcting grammatical errors, and refining academic expression to meet the journal’s standards.

Data Presentation: I recommend a review of how figures and tables are integrated into the paper. They should be clearly labeled and directly referenced in the text, ensuring they effectively support the paper's conclusions.

Reply: We have systematically reviewed the manuscript data to identify and eliminate potential errors, including data entry inconsistencies, statistical calculation discrepancies, and figure caption mismatches.

Logical Flow: Ensuring a coherent flow throughout the paper, with seamless transitions between sections, is key to keeping the reader engaged and ensuring the paper is easily understandable.

Reply: We have reviewed and checked the entire text.

Abstract and Conclusion: These sections should succinctly summarize the main findings and their significance. It is important to avoid overstating results or generalizations not supported by the data.

Reply: We have reviewed and confirmed the abstract and conclusions.

Consistency in Terminology: Uniform use of technical terms throughout the manuscript is essential for clarity and to avoid potential confusion among readers.

Reply: We have reviewed and checked the entire text.

Objective Language: Maintaining an objective tone throughout the paper is crucial. Subjective or evaluative language should be reserved for discussing hypotheses or theoretical frameworks.

Reply: We have reviewed and checked the entire text.

Compliance with Journal Guidelines: Finally, ensuring that the manuscript aligns with the journal's specific guidelines in terms of formatting, submission, and ethics will facilitate a smoother review and publication process.

Reply: We have checked the manuscript against the journal guidelines.

I trust these suggestions will be valuable in enhancing the manuscript's effectiveness and suitability for publication. Thank you for the opportunity to review this work.

---

## [Decision Letter · Decision Letter 1]

9 Dec 2025

Dear Dr. Liu,

We look forward to receiving your revised manuscript.

Kind regards,

Jianhong Zhou

Staff Editor

PLOS One

Saki Raheem, PhD

Academic Editor

PLOS ONE

Journal Requirements:

Reviewers' comments:

Reviewer's Responses to Questions

**Comments to the Author**

Reviewer #1: All comments have been addressed

2. Is the manuscript technically sound, and do the data support the conclusions?

Reviewer #1: Yes

3. Has the statistical analysis been performed appropriately and rigorously?

Reviewer #1: I Don't Know

4. Have the authors made all data underlying the findings in their manuscript fully available?

Reviewer #1: Yes

5. Is the manuscript presented in an intelligible fashion and written in standard English?

Reviewer #1: Yes

Reviewer #1: I am writing to provide feedback on the manuscript titled "RIPK2 induces docetaxel resistance in prostate cancer through the NF-κB/P-gp signaling pathway" which I understand is under consideration for publication.

The comments raised have been corrected and are acceptable.

**Do you want your identity to be public for this peer review?** For information about this choice, including consent withdrawal, please see our Privacy Policy

Reviewer #1: **Yes:** Mohsen Rashidi

---

## [Author Response · Author response to Decision Letter 2]

18 Dec 2025

2025/12/11

Dear Editor:

We thank the editors and reviewers for all their efforts in reviewing the manuscript. In response to the comments raised in the second round of review, we hereby provide detailed data regarding the maximum tumor size in our xenograft experiments. None of the xenograft tumors exceeded 10% of the host mice’s body weight, which is in strict compliance with the ethical guidelines for animal experiments. The largest tumor, with a volume of 1991.23 mm³ and a weight of 1.321 g, accounted for only 4.98% of the corresponding mouse’s body weight (26.5 g). It should be noted that tumor volume measurements might be slightly overestimated due to the irregular morphology of xenograft tumors; however, the overall results are consistent with our experimental expectations.

Thank you for your consideration. I look forward to hearing from you.

Sincerely,

Qian Liu

Department of Urology, First Central Clinical College, Tianjin Medical University, No. 24, Fukang Road, Nankai District, Tianjin, 300190, China. +86 18963106651

simonlq1971@126.com

---

## [Editor Report · Decision Letter 2]

22 Dec 2025

Dear Dr. Liu,

Thank you for submitting your manuscript to PLOS ONE. After careful consideration, we feel that it has merit but does not fully meet PLOS ONE’s publication criteria as it currently stands. Therefore, we invite you to submit a revised version of the manuscript that addresses the points raised during the review process.

We look forward to receiving your revised manuscript.

Kind regards,

Miquel Vall-llosera Camps

Senior Staff Editor

PLOS One

Saki Raheem, PhD

Academic Editor

PLOS ONE

**Journal Requirements:**

**Additional Comments from the Editorial Staff:**

Thank you for your response to our previous requests. At this time, please provide the complete data points underlying Figure 6C, to confirm that 1991.23 being the largest tumour size is correct, given that the error bars appear to indicate otherwise.

---

## [Author Response · Author response to Decision Letter 3]

23 Dec 2025

2025/12/23

Dear Editor:

We thank the editors and reviewers for all their efforts in reviewing the manuscript. In response to the comments raised in the second round of review, we hereby provide detailed data regarding the maximum tumor size in our xenograft experiments. None of the xenograft tumors exceeded 10% of the host mice’s body weight, which is in strict compliance with the ethical guidelines for animal experiments. The largest tumor, with a volume of 1991.23 mm³ and a weight of 1.321 g, accounted for only 4.98% of the corresponding mouse’s body weight (26.5 g). It should be noted that tumor volume measurements might be slightly overestimated due to the irregular morphology of xenograft tumors; however, the overall results are consistent with our experimental expectations.

We have uploaded the detailed data on tumor volumes in each group (Day 28) under the “Other” section. Please refer to this dataset.

Thank you for your consideration. I look forward to hearing from you.

Sincerely,

Qian Liu

Department of Urology, First Central Clinical College, Tianjin Medical University, No. 24, Fukang Road, Nankai District, Tianjin, 300190, China. +86 18963106651

simonlq1971@126.com

---

## [Editor Report · Decision Letter 3]

7 Jan 2026

RIPK2 induces docetaxel resistance in prostate cancer through the NF-κB/P-gp signaling pathway

PONE-D-25-24339R3

Dear Dr. Liu,

We’re pleased to inform you that your manuscript has been judged scientifically suitable for publication and will be formally accepted for publication once it meets all outstanding technical requirements.

Kind regards,

Saki Raheem, PhD

Academic Editor

PLOS One

---

## [Editor Report · Acceptance letter]

PONE-D-25-24339R3

PLOS One

Dear Dr. Liu,

I'm pleased to inform you that your manuscript has been deemed suitable for publication in PLOS One. Congratulations! Your manuscript is now being handed over to our production team.

Kind regards,

on behalf of

Dr. Saki Raheem

Academic Editor

PLOS One